# Exploring the Relationship Between Health and Out-of-Pocket Health Expenditures: Evidence for Middle-Aged and Older Adults in China

**DOI:** 10.3390/healthcare12212137

**Published:** 2024-10-27

**Authors:** Jingyi Gao

**Affiliations:** Department of Economics, Fordham University, Bronx, NY 10458, USA; jgao44@fordham.edu

**Keywords:** disability, chronic conditions, out-of-pocket expenditures, healthcare, social health insurance

## Abstract

Background/Objectives: With population aging, disability and chronic conditions are increasingly prevalent among middle-aged and older adults in China. Using panel data from the China Health and Retirement Longitudinal Study (CHARLS) from 2011 to 2018, this paper explores the effects of Activities of Daily Living limitations, Instrumental Activities of Daily Living limitations, and chronic conditions on Out-of-Pocket Expenditures (OOPEs) among middle-aged and older adults in China. Methods: A first-difference model and a system-generalized method of moment model (GMM) are used. Results: The system-GMM model for the first time addresses unobserved heterogeneity and produces unbiased estimates of the effects of health and OOPEs. Additionally, this paper assesses the heterogeneity of the results across the demographic and socioeconomic groups. Conclusions: These findings can be used to inform policymakers on improving medical resource allocation and ensure better financial protection for those living with a disability and chronic diseases.

## 1. Introduction

The aging population is becoming a significant global concern, with profound implications for many countries, including China. By 2030, one in six people worldwide will be aged 60 years or over, and the number of individuals in this age group is projected to grow from 1 billion in 2020 to 1.4 billion. By 2050, the global population of those aged 60 years and older is expected to double, reaching 2.1 billion [1]. China, in particular, is experiencing a rapid increase in its aging population. The older population (aged 60 years or older) reached 18.7% of the population according to the latest census data update in 2021 [2]. A high prevalence of disability and chronic conditions is associated with aging. For instance, around 75% of the older population experience chronic conditions including diabetes, hypertension, and other chronic conditions [3]. It is found that 26.2% of the older population in China suffer from disability based on the meta-analysis of 97 studies from 1979 to 2022 [4]. The high prevalence of disability and chronic conditions among the old presents challenges to the public health system in China.

Aiming to provide universal financial protection to individuals and achieve universal health coverage, in 1998, the Chinese government established a series of social health insurance schemes targeting both urban residents at the municipal level and rural residents at the country level: the Urban Employee Basic Medical Insurance (UEBMI), Urban Resident Basic Medical Insurance (URBMI), and the New Rural Cooperative Medical Scheme (NRCMS) [5]. In April 2009, the government established another round of comprehensive health system reforms involving medical care services policy to provide a universal healthcare system [6]. Out-of-pocket expenditures (OOPEs) decreased from 56% in 2003 to 29% in 2017 as the social health insurance coverage expanded. There has also been greater health service utilization [7]. However, problems of inequality in healthcare and insufficiency of cost coverage remain in socioeconomically underdeveloped regions of China [8]. It is imperative to provide an in-depth analysis of healthcare expenditures in different population groups by demographics, geography, and income level to inform policies that allocate resources to relieve healthcare access disparities and the burden of OOPEs on households.

There is an extensive literature examining the association between higher out-of-pocket expenditures (OOPEs) and health status. However, previous studies have not addressed the issues of unobserved heterogeneity and measurement error in these analyses. Additionally, there is a limited understanding of how OOPEs are influenced specifically by activity limitations and chronic conditions across different demographic subgroups in China. This research paper fills this gap by generating unbiased and consistent estimates of the effects of health status, as measured by activity limitations and chronic diseases, on OOPEs among middle-aged and elderly adults in China. It also explores whether the impacts of activity limitations and chronic conditions on OOPEs vary across different subgroups including gender, income, and Hukou/residence arrangement, potentially leading to targeted health policy interventions. The following sections review the relevant literature, describe the dataset, measures and models, and results. The last Section concludes.

## 2. Literature Review

Many studies have estimated the healthcare expenditure and extra cost of disability across demographic and socioeconomic groups. Owens (2008) analyze the gender differences in healthcare expenditures and resource utilization in the United States. He finds that healthcare expenditures tend to be higher among women than among men. The greatest disparity in healthcare spending between men and women occurred in the population aged 45 to 64 years with women’s health issues primarily revolving around chronic conditions and menopausal symptoms in this age group [9]. Loyalka et al. (2014) measure the extra cost of disability across different types of disability and households in both urban and rural areas of China. They find the extra costs of disability are larger for urban households than for rural households, and there is a strong negative correlation between disability and household income [10]. Lee et al. (2014) examine medical expenditures among non-institutionalized adults in the United States with one or more chronic conditions using data from the 2010 Medical Expenditure Panel Survey Household Component. This study provides descriptive statistics on the high healthcare expenditures associated with age, race/ethnicity, marital status, insurance status, and education and concludes that individuals with chronic conditions experienced higher total spending than those with no chronic conditions after controlling for confounding factors [11]. Lee et al. (2016) investigate the burden of catastrophic health expenditures on households with disabled members compared to those without, utilizing the Korean Health Panel data from 2010 and 2011. It found that households with a disabled member spend 1.2 to 1.4 times more on medical expenses than those without disabled members, which highlights the need for policy adjustments to enhance financial support to aid disabled individuals experiencing financial hardship in accessing necessary medical services [12]. 

There have been many studies that examine the relationship between OOPEs and health problems. They use cross-sectional data and estimate a correlation. Zhao et al. (2021) use quantile regressions to identify the positive association between multimorbidity and OOPEs in China using CHARLS 2015 data [13]. You and Kobayashi (2011) use 2004 China Health and Nutrition Survey data and apply the Heckman selection model and find that OOPEs are higher for individuals who had chronic conditions, earned higher income, resided in urban areas, lived in the middle or eastern region, or lived in a household with a head having a middle school or higher education. They conclude that the perceived severity of illness and self-reported health status are the most important determinants of OOPEs [14]. Salinas-Rodriguez et al. (2020) find a positive association between activity of daily living (ADL)/instrumental activity of daily living (IADL) limitations and OOPEs using a two-part regression model and quantile regression among the Mexican adults aged 60 and older [15]. Nguyen et al. (2021) also apply the two-part regression models to find that disability is strongly associated with higher OOPEs in Vietnam [16]. Gao et al. (2023) apply both ordinary least squares linear regression and logistic regression models to analyze OOPEs associated with chronic health conditions and disabilities in China, utilizing data from CHARLS 2018. The findings indicate a need for increased policy and research focus on enhancing financial protection for individuals with chronic conditions and disabilities, particularly through expanded access to comprehensive health insurance coverage [17]. These studies focus on the correlations between higher OOPEs and health conditions, but they do not address the unobserved heterogeneity and examine the unbiased effects of health on OOPEs. 

A growing number of studies address the unobserved heterogeneity and examine the unbiased consistent effects of health on economic indicators using a dynamic panel data model to overcome the limitations of the static models. However, these studies do not use OOPEs as an outcome of interest. Kim and Mitra (2022) examine the two-way causal relationships between health and labor income among middle-aged and older Koreans. They apply the dynamic panel data model with 12 waves of data from 2006 to 2017. They also stratify the sample based on age, gender, region, income level, and marital status to analyze the outcomes across different demographic and socioeconomic groups [18]. Meraya et al. (2017) examine the dynamic relationships between economic status and health using eight waves of panel data. A causal link between labor income and self-rated health and functional status for both genders is found using a system-generalized method of moment [19]. Our study follows the dynamic panel data model framework addressing the unobserved heterogeneity, and for the first time provides unbiased estimates of the effects of health on OOPEs among middle-aged and older adults in China using four waves of CHARLS from 2011 to 2018.

## 3. Data and Methods

This paper uses the 2011, 2013, 2015, and 2018 waves of the China Health and Retirement Longitudinal Study (CHARLS), which is a nationally representative longitudinal survey of people aged 45 and over and their partners in China. The CHARLS datasets provide demographic, socioeconomic, and health information at the individual and household levels. The unit of analysis is the individual. Individual cross-sectional weights with household and individual non-response adjustments are applied to all results.

The baseline survey for CHARLS was conducted in 2011/2012 and included 10,257 households and 17,500 individual respondents. The survey collects information on the demographics, family, health status, healthcare, health insurance, and wealth at both individual and household levels. Specifically, CHARLS has a series of questions on OOPEs. All respondents are asked to report how many times they have received outpatient care and inpatient care in the past, and the total medical cost of all the doctor visits and hospitalizations. 

Since ADL/IADL-related questions were not asked of respondents aged 45–49, this study focuses on 7448 respondents aged 50 and over who can be followed over the four waves from 2011 to 2018, forming a balanced panel dataset after removing missing values. And only respondents covered by public health insurance (UEBMI, URBMI, and NRCMS) are included in this study. 

### 3.1. Hypothesis

As individuals experience health issues such as activity limitations and chronic conditions, healthcare services are expected to be used more often, which will lead to OOPEs. Thus, the first hypothesis is that health problems have a positive effect on OOPEs. 

How health is measured could influence the magnitude of this effect. For ADL/IADL limitations, respondents are asked whether they have any difficulty performing activities of daily living (e.g., bathing), and ADL/IADL indices are constructed from their answers. Answers are intuitive and do not require the respondents to have used the healthcare system, but persons who have used healthcare services may be more aware of their limitations. The chronic condition index is constructed from individuals answering about whether they have been diagnosed by a doctor for specific health conditions (e.g., diabetes). This requires that individuals have used the healthcare system to see a doctor and have potentially been given a diagnosis, and, in this study, they are considered to have multimorbidity. Using the healthcare system generates more OOPEs and may lead to a diagnosis. The second hypothesis is that multimorbidity leads to more OOPEs than ADL or IADL limitations. 

In addition, self-reported health indicators, as used in this study, may be biased by reporting errors. Health self-reports can be affected by socioeconomic characteristics such as gender, education, income, and so on [20,21,22]. For instance, high-income individuals tend to use healthcare services more than low-income individuals. They thus incur more OOPEs and are more aware of their health issues. The third hypothesis is that the effect of health on OOPEs is larger for more advantaged socioeconomic groups (e.g., high-income, more highly educated).

### 3.2. Out-of-Pocket Expenditures (OOPEs)

OOPEs are the expenses for both outpatient and inpatient visits that are not covered by health insurance. Outpatient OOPEs are measured by the value of out-of-pocket doctor-visit expenditures for the respondent in the last month covering the fees paid for treatment, medication costs, and prescription drugs. Inpatient OOPEs are measured by the value of out-of-pocket hospitalization expenditure for the respondent in the past year. As out-of-pocket doctor-visit expenditure in CHARLS is measured for the last month only, the value is multiplied by 12 to estimate annual outpatient expenditures. OOPEs, the target of interest, are thus the sum of outpatient OOPEs and inpatient OOPEs.

### 3.3. Health Measures

Health measures cover disability and chronic health conditions. Disability is measured by ADL and IADL limitations. To measure the severity of ADL or IADL limitations, ADL and IADL indices developed by Stewart and Ware (1992, p. 80) are used, which have been widely used in the literature subsequently (e.g., Gertler and Gruber, 2002) [23,24]. For ADL limitations, each respondent is asked whether they experience some difficulty performing the designated six tasks including bathing, dressing, eating, getting in/out of bed, using the toilet, and controlling urination. For IADL limitations, each respondent is asked whether they experience some difficulty performing the designated five tasks including using the phone, managing money, taking medications, shopping for groceries, and preparing hot meals. A zero indicates that the respondent did not report any problem with each activity. A one indicates that the respondent reported some difficulty or could not do each activity. To capture the number of limitations, an ADL sum/IADL sum is used: the ADL sum (from 0 to 6)/IADL sum (from 0 to 5) is the number of yes answers, respectively. Both the ADL and IADL sums are normalized to a scale of 100 to form the ADL index and the IADL index, respectively.

In addition, a measure of chronic health conditions that captures 15 chronic conditions (hypertension, diabetes, cancer, chronic lung disease, heart disease, stroke, psychiatric problems, arthritis, dyslipidemia, liver disease, kidney disease, digestive disease, asthma, depression, and memory problem) is used. Each respondent answered the question regarding whether or not they have been diagnosed by the doctor for any of the conditions above. Then, the number of diagnosed chronic conditions for each respondent is counted. A Non-Communicable Disease (NCD) index is constructed by normalizing the count to 100. Multimorbidity is defined as the presence of two or more chronic diseases and is measured by a binary variable coded as 1 for respondents with multimorbidity and 0 otherwise. 

### 3.4. Models

The objective is to identify the effect of health on OOPEs, and this is completed in several steps, which start with the first-difference model. The first-difference estimator addresses the issue of individual fixed effects by using the changes between two time periods for each individual. This approach effectively eliminates the fixed individual-specific effects, as these do not vary over time. The model specification for this estimator is as follows:(1)$${∆Y}_{it}=\beta_{1}{∆H}_{it}+\sum_{j=2}^{R}\beta_{j}{∆X}_{jit}+\Delta \mu_{it}$$
where i indexes individuals and t indexes time periods. ${∆Y}_{it}$ is the change in the log-transformed of $Y_{it}$, which is the individual’s OOPEs in US dollars at t. ${∆H}_{it}$ is the change in health for individual i between time t and time t − 1. $H_{it}$ is measured in turn through an ADL Index, IADL Index, and NCD Index at t. $\beta_{1}$ is the main coefficient of interest. ${∆X}_{jit}$ represents the change in the time-varying control variables at the individual level including marriage, Hukou/living arrangement, household size, and moving status (moved outside the original community of 2011). Time-invariant controls (gender, education) are removed by using the first-difference estimator because they are time-invariant. Time-invariant unobservables $\varepsilon_{i}$ are also differenced out.

The term $\Delta \mu_{it}$ is the change in the individual-specific error term that accounts for the change in time-varying unobservables for the respondent. 

The first-difference model above captures the extent to which changes in OOPEs are associated with changes in health. While it can handle the problem of unobserved time-invariant heterogeneity, it will not resolve the issue of omitted variables bias due to the presence of time-varying unobservables. It also does not address the time-dependence of OOPEs [25].

As current OOPEs may be influenced by past OOPEs and current health and other variables, another approach is to model the dynamic relationship between OOPEs and health as follows:(2)$$Y_{it}={\beta_{0}Y_{it-1}+\beta }_{1}H_{it}+\sum_{j=2}^{R}\beta_{j}X_{jit}+\varepsilon_{i}+\mu_{it}$$

To address time-invariant heterogeneity, Equation (3) can be first differenced as follows:(3)$${∆Y}_{it}={\beta_{0}{∆Y}_{it-1}+\beta }_{1}∆H_{it}+\sum_{j=2}^{R}\beta_{j}{∆X}_{jit}+{∆\mu }_{it}$$

Equation (3) addresses unobserved time-invariant heterogeneity. However, the first-differenced lagged dependent variable could be correlated with the first-differenced error term. An Arellano and Bond (1991) estimator can address this issue: the second and third lags of the dependent variable are used as instruments for the first-differenced lagged dependent variable. Those lags will be highly correlated with the first-differenced lagged dependent variable but uncorrelated with the composite error process [26]. However, this approach may encounter a weak instrument issue, as the lagged values of endogenous variables might weakly correlate with the regressors in the first-differenced model. Addressing this concern, the Blundell and Bond (1998) system-GMM estimator is applied, which imposes stricter moment conditions. The system-GMM estimator uses lagged differences as instruments for the level model and lagged levels as instruments for the first-differenced model [27].

The utilization of system-GMM in the estimation has three main issues: the instrumental lag selection, the proliferation of instruments, and the serial autocorrelation of errors. Firstly, in this case, only four waves of panel data are available, and the second and third lags of the dependent variable, as well as up to the third lagged difference, are the only choices for instruments. Secondly, a crucial assumption for the validity of GMM is that the instruments are exogenous. If the model is exactly identified, the detection of invalid instruments is impossible. But if the model is overidentified, a test statistic for the joint validity of the moment conditions (identifying restrictions) falls naturally out of the GMM framework. The existence of excess of instruments could be tested through the Sargan and Hansen tests [28]. Thirdly, since only four waves of panel data are available, the assumption of no second-order serial correlation of the error term cannot be tested. 

## 4. Results

### 4.1. Descriptive Statistics

The descriptive statistics are analyzed on 7448 respondents aged 50 and over. Table 1 presents the mean OOPEs in USD and the presence of any OOPEs stratified by groups of demographics including gender, income, and Hukou/residence arrangements. The average OOPE is USD 396.51 among men, while the average OOPE is USD 461.66 among women. Women experience both a higher occurrence of OOPE and higher OOPEs. Our sample is also stratified into four income groups. The average OOPE among the high-income and low-income is higher than that among the middle-income group based on our statistical results. Moreover, higher average OOPEs appear in the groups with urban Hukou and lower average OOPEs appear in the groups with rural Hukou, where Hukou is a household registration system that categorizes each Chinese citizen as either an agricultural (rural) Hukou holder or a non-agricultural (urban) Hukou holder.

Table 2 focuses on individuals across all four waves and provides the percentage of individuals with strictly positive OOPEs along with the average OOPEs. These figures are categorized based on individuals’ ADL limitations, IADL limitations, and multimorbidity status. On average, individuals spend USD 429.71 per year on OOPEs, with 26.64% of them incurring strictly positive OOPEs. Those with ADL limitations, IADL limitations, or multimorbidity are more likely to have strictly positive OOPEs, and their average expenses are higher. For example, individuals with IADL limitations spend an average of USD 843.41 on OOPEs, compared to USD 324.33 for those without IADL limitations.

Table 3 displays the prevalence of ADL limitation, IADL limitation, and multimorbidity (expressed as percentages) by sociodemographic characteristics of the respondents. As shown from the table, the presence of disability measured by ADL limitation and IADL limitation increases as the age group increases, while the presence of multimorbidity is relatively stable across the age groups. Moreover, the presence of a disability is lower in the above-median income group compared to the below-median income group, while this disparity is not detected for multimorbidity. The prevalence of disability is lowest in the urban Hukou/urban residence arrangement. It can also be concluded that the presence of ADL limitation/IADL limitation/multimorbidity is higher among women and individuals who are not married.

### 4.2. Model Results 

Results of the models (1) and (3) are presented below. To assess the potential heterogeneity of the results across different groups, for each model, our sample is stratified by gender, income (highest income quantile and lowest income quantile), and Hukou/residence (rural Hukou/rural residence, rural Hukou/urban residence, urban Hukou/urban residence, urban Hukou/rural residence).

The results of the first-difference regression (Equation (1)) are displayed in Table 4a,b. A positive relationship between health, whatever the measure, and the natural logarithm of OOPEs is found. It can be concluded that the association is larger between health and OOPEs for the high-income group compared to the low-income group. For instance, the estimation results suggest that for the high-income group, a ten-unit increase in the NCD index raises OOPEs by 63%, while for the low-income group, a ten-unit increase in the NCD index raises OOPEs by 36%. Moreover, the largest association between health and OOPEs is found for urban Hukou/urban residence.

The results of the system-GMM are in Table 5a,b. A statistically significant positive relationship between health and OOPEs is found. A 10-unit increase in ADL index raises OOPEs by 16% for women, and by 18% for men. A 10-unit increase in the IADL index raises OOPEs by 15% for women, and by 18% for men. A rise in the NCD index leads to a higher increase in OOPEs than a rise in the ADL or IADL index for both men and women. 

Next, the sample is stratified by income level. The same unit increase in health results in a larger increase in OOPEs for the high-income group than for the low-income group. For instance, a 10-unit increase in ADL index raises OOPEs by 27% among the high-income group, and by 16% among the low-income group. Similarly, the increase in the NCD index leads to a higher increase in OOPEs compared to the ADL/IADL index among both the high-income group and the low-income group.

The sample is further stratified into four Hukou/residence arrangements. It is found that the increase in health measures results in a larger increase in OOPEs for the Urban Hukou/Urban Residence group compared with the Rural Hukou/Rural Residence group. For instance, a 10-unit increase in ADL index/IADL index/NCD index increases OOPEs by 34%, 33%, and 45%, respectively, among the Urban Hukou/Urban Residence group, while a 10-unit increase in ADL index/IADL index/NCD index increases OOPEs by 15%, 12%, and 30%, respectively, among the Rural Hukou/Rural Residence group. 

## 5. Discussion

A positive relationship between health problems and OOPEs was found from both first-difference and Blundell–Bond estimates. This result is consistent with earlier studies on the associations between health and OOPEs [13,15,16]. Another finding is that the increase in the Non-Communicable Disease (NCD) index leads to a larger increase in OOPEs compared to the ADL/IADL index. This result is in accordance with our hypothesis that the ADL/IADL index is constructed by individual self-report and the NCD index is constructed by self-report from doctor diagnosis. The self-reported index and doctor-diagnosed index give different results. Individuals are using and paying for health services when diagnosed with chronic diseases by a doctor. But for self-reported disability, individuals describe their disability by themselves directly and do not need to go through the doctor diagnostic process. 

The sample is stratified by subgroups and generates heterogeneous results across the demographic subgroups. It is found from the descriptive statistics that average OOPEs are higher for female, high-income, and urban Hukou/urban residence subgroups. Based on the model estimations, the increase in health measures leads to higher OOPEs among the high-income group and urban Hukou/urban residence arrangement. These results are consistent with earlier studies. Women tend to use significantly more services and spend more healthcare dollars than men due to the onset of menopausal symptoms in middle age [9]. You and Kobayashi (2011) find that individuals who earned higher incomes and resided in urban areas incurred more OOPEs [14]. Loyalka et al. (2014) also talk about the urban–rural disparity in the extra costs of disability in China. They find the extra costs of disability are larger for urban households than rural households since the accessibility to healthcare services for the disabled for rural households is less than that for urban households and rural households are in poorer condition to purchase the services compared to urban households. Better accessibility to health services improves the self-awareness of health status and increases the incidence of diagnoses of health issues, leading to higher OOPEs [10].

However, this study has several limitations. The system-GMM estimator uses lagged differences as instruments for the level model and lagged levels as instruments for the first-differenced model. Only four waves of panel data are available at the time of the study, which provides limited choices of instrument variables. Moreover, OOPEs and health are explored in a broad sense due to data availability. Further research is needed on how different types of health insurance coverage may affect OOPEs for persons with disability and chronic health conditions. 

This study makes several contributions. It is the first study to address the unobserved heterogeneity and examine the unbiased consistent effects of ADL/IADL limitations and chronic conditions on OOPEs in China. Positive effects from health on OOPEs have been identified, which provides novel evidence on the financial burden of ADL/IADL limitations and chronic conditions among the aging group in China. Second, this paper assesses the heterogeneity of the OOPEs across demographic and socioeconomic groups. The results imply the poorer accessibility of healthcare services, especially among the most socioeconomically disadvantaged group. These findings point out a need for policymakers to increase healthcare subsidies, expand and tailor health insurance coverage, enhance community-based health planning, initiate health education campaigns, etc. Additionally, improving medical resource allocation in under-served areas, incentivizing healthcare professionals to work in rural regions, and implementing robust monitoring and evaluation systems are critical. These measures could ensure equitable healthcare access across all socioeconomic groups, particularly for low-income, rural residents.

Finally, this paper addresses the different effects of chronic conditions and activity limitations on OOPEs. The association of chronic conditions with OOPEs is larger than that of ADL or IADL limitations. When interpreting the effects of health on OOPEs, random measurement error, unobserved time-varying heterogeneity, and reverse causality from OOPEs to health need to be taken into account. Specifically, people who have higher OOPEs use health services more often, and thus are more likely to have diagnoses and report chronic health conditions. For ADL/IADL limitations, it may work differently: people who have higher OOPEs and use health services are more likely to be aware of their ADL/IADL limitations. Thus, the two-way causality linking health and OOPEs needs to be examined further.

## 6. Conclusions

This study, for the first time, addresses unobserved heterogeneity and provides unbiased estimates of the effects of ADL/IADL limitations and chronic conditions on OOPEs in China, revealing significant financial burdens on the aging population and disparities in healthcare access among socioeconomically disadvantaged groups. These findings underscore the need for policymakers to improve medical resource allocation, particularly in rural and low-income areas, and to strengthen financial protection mechanisms through enhanced health insurance and financial assistance programs. Moreover, the study suggests tailoring health policies to demographic variations, such as gender, income, and residence arrangements, to address disparities and reduce OOPEs, thereby promoting health equity and financial protection for vulnerable populations. Additionally, to mitigate the challenges posed by an aging population, policymakers should also focus on developing comprehensive long-term care systems, increasing support for caregivers, investing in preventive healthcare to reduce the onset of chronic conditions, and expanding community-based services that allow the aging group to maintain independence and reduce reliance on expensive healthcare services. 

## Figures and Tables

**Table 1 healthcare-12-02137-t001:** Out-of-pocket health expenditures, stratified by gender, income, and Hukou/residence arrangement.

Group	Features	OOPEs				Share with OOPEs	Sample Size
		Mean	Std Dev	Min	Max		
Gender	Male	396.51	2602.23	0	120,860	0.251	3653
	Female	461.66	3114.01	0	143,312	0.281	3795
Income	Low	454.02	3505.76	0	143,312	0.277	1867
	Low-middle	385.75	2415.19	0	103,503	0.279	1879
	Middle-high	411.50	2502.65	0	76,433	0.257	1840
	High	467.68	2941.43	0	120,860	0.253	1862
Hukou/Residence	Rural Hukou/Rural Residence	399.17	2896.81	0	143,312	0.260	4648
	Rural Hukou/Urban Residence	386.45	2168.72	0	76,433	0.269	1379
	Urban Hukou/Urban Residence	597.05	3397.75	0	120,860	0.283	1080
	Urban Hukou/Rural Residence	505.55	3586.47	0	101,274	0.322	226

Notes: OOPEs stand for out-of-pocket health expenditures. OOPES are in USD (reference year 2015); Share with OOPEs refers to the proportion of respondents experiencing positive OOPEs. Source: Author’s calculations based on 2011–2018 CHARLS data.

**Table 2 healthcare-12-02137-t002:** Mean out-of-pocket health expenditures by ADL limitation/IADL limitation/Multimorbidity status.

	All	ADL Limitation			IADL Limitation			Multimorbidity		
		Yes	No	Diff.	Yes	No	Diff.	Yes	No	Diff.
**n**	29,792	6048	23,744		7216	22,576		16,109	13,683	
**Mean of OOPEs (USD)**	429.71 (2874)	843.41 (4526)	324.33 (2257)	519.08 *** [0.00]	847.99 (4589)	296.01 (2025)	551.98 *** [0.00]	631.36 (3688)	192.29 (1370)	439.07 *** [0.00]
**%Incurring OOPEs**	26.64 (44.21)	39.24 (48.83)	23.44 (42.36)	15.80	36.95 (48.27)	23.35 (42.31)	13.60	34.97 (47.69)	16.85 (37.43)	18.12

Notes: *** *p* < 0.01, ** *p* < 0.05, * *p* < 0.1; Standard deviation in parentheses and *p* values in bracket. Source: Authors’ calculations based on 2011–2018 CHARLS data.

**Table 3 healthcare-12-02137-t003:** Descriptive statistics (in percentages).

	Full Sample (n = 7488)	Persons with ADL Limitation	Persons with IADL Limitation	Persons with Multimorbidity
**Age (year)**				
50–59	50.3	15.3	18.9	53.0
60–69	36.0	22.4	26.0	60.8
70 and above	13.7	33.2	39.1	59.2
**Gender**				
Male	49.0	16.3	18.7	51.3
Female	51.0	24.2	29.5	61.8
**Marital status**				
Married and partnered	85.4	18.8	22.6	55.6
Unmarried and others	14.6	28.8	33.7	61.8
**Education**				
Illiterate	50.1	25.1	31.4	59.2
Primary school	24.2	12.8	13.6	51.5
Secondary school	22.3	19.0	21.5	56.0
College and above	3.3	11.8	11.4	60.0
**Income**				
Low	25.0	28.2	28.8	59.0
High	25.0	11.7	12.7	52.2
**Hukou/Residence**				
Rural Hukou/Rural Residence	62.4	22.3	26.8	56.1
Rural Hukou/Urban Residence	18.5	18.5	22.4	55.3
Urban Hukou/Urban Residence	14.5	14.4	17.1	59.8
Urban Hukou/Rural Residence	3.0	20.9	19.7	59.6

Source: Author’s calculations based on 2011–2018 CHARLS data.

**Table 4 healthcare-12-02137-t004:** (**a**) First-difference regression of the effects of health on the natural logarithm of OOPEs, stratified by gender and income. (**b**) First-difference regression of the effects of health on the natural logarithm of OOPEs, stratified by Hukou/residence.

**(a)**				
	**Male**	**Female**	**Low Income**	**High Income**
VARIABLES	(1)	(2)	(3)	(4)
ADL index	0.019 ***	0.022 ***	0.019 ***	0.021 ***
	(0.002)	(0.002)	(0.003)	(0.004)
IADL index	0.017 ***	0.015 ***	0.015 ***	0.023 ***
	(0.002)	(0.002)	(0.002)	(0.004)
NCD index	0.060 ***	0.037 ***	0.036 ***	0.063 ***
	(0.005)	(0.005)	(0.007)	(0.004)
Observations	11,072	11,116	5508	5544
**(b)**				
	**Rural Hukou/Rural Residence**	**Rural Hukou/Urban Residence**	**Urban Hukou/Urban Residence**	**Urban Hukou/Rural Residence**
VARIABLES	(5)	(6)	(7)	(8)
ADL index	0.016 ***	0.019 ***	0.025 ***	0.018 *
	(0.002)	(0.002)	(0.003)	(0.004)
IADL index	0.012 ***	0.020 ***	0.027 ***	0.011
	(0.002)	(0.002)	(0.002)	(0.004)
NCD index	0.041 ***	0.040 ***	0.060 ***	0.099 ***
	(0.005)	(0.005)	(0.007)	(0.004)
Observations	18,840	5620	4420	720

Notes: *** *p* < 0.01, ** *p* < 0.05, * *p* < 0.1. Robust standard errors are presented in parentheses. Regressions are adjusted for survey weights. The vector of controls includes marriage, Hukou/living arrangement, household size, and moving status (moved outside the community in 2011). Source: Author’s calculations based on 2011–2018 CHARLS data.

**Table 5 healthcare-12-02137-t005:** (**a**) Blundell–Bond estimates of the effects of health on the natural logarithm of OOPEs, stratified by gender and income. (**b**) Blundell–Bond estimates of the effects of health on the natural logarithm of OOPEs, stratified by Hukou/residence.

**(a)**				
	**Male**	**Female**	**Low Income**	**High Income**
VARIABLES	(1)	(2)	(3)	(4)
ADL index	0.018 ***	0.016 ***	0.016 ***	0.027 ***
	(0.000)	(0.000)	(0.000)	(0.000)
Lagged OOPE	0.039*	0.031	0.044	0.040
	(0.066)	(0.136)	(0.127)	(0.191)
IADL index	0.018 ***	0.015 ***	0.013 ***	0.028 ***
	(0.000)	(0.000)	(0.000)	(0.000)
Lagged OOPE	0.037 *	0.030	0.039	0.039
	(0.083)	(0.141)	(0.171)	(0.197)
NCD index	0.047 **	0.024 ***	0.025 ***	0.053 ***
	(0.000)	(0.000)	(0.003)	(0.000)
Lagged OOPE	0.043 **	0.033	0.044	0.039
	(0.043)	(0.111)	(0.124)	(0.197)
Observations	5536	5558	2754	2772
**(b)**				
	**Rural Hukou/Rural Residence**	**Rural Hukou/Urban Residence**	**Urban Hukou/Urban Residence**	**Urban Hukou/Rural Residence**
VARIABLES	(5)	(6)	(7)	(8)
ADL index	0.015 ***	0.015 ***	0.034 ***	0.026 *
	(0.000)	(0.003)	(0.000)	(0.073)
Lagged OOPE	0.043 **	−0.020	0.092 **	0.012
	(0.023)	(0.549)	(0.026)	(0.875)
IADL index	0.012 ***	0.022 ***	0.033 ***	0.019*
	(0.000)	(0.000)	(0.000)	(0.090)
Lagged OOPE	0.042 **	−0.021	0.092 **	0.011
	(0.028)	(0.527)	(0.025)	(0.891)
NCD index	0.030 *** (0.000)	0.030 *** (0.003)	0.045 *** (0.001)	0.101 *** (0.000)
Lagged OOPE	0.047 **	−0.017	0.090 **	−0.009
	(0.013)	(0.614)	(0.031)	(0.900)
Observations	18,840	5620	4420	720

Notes: *** *p* < 0.01, ** *p* < 0.05, * *p* < 0.1. Robust standard errors are presented in parentheses. The vector of controls includes marriage, Hukou/living arrangement, household size, and moving status (moved outside the community in 2011). Source: Author’s calculations based on 2011–2018 CHARLS data.

## Data Availability

Publicly available datasets were analyzed in this study. These data can be found here: [https://charls.charlsdata.com/pages/Data/harmonized_charls/en.html] (accessed on 1 November 2022).

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
