# Peer review of "Exploring the Relationship Between Health and Out-of-Pocket Health Expenditures: Evidence for Middle-Aged and Older Adults in China"

_healthcare, 2024, doi:10.3390/healthcare12212137_

Round 1
Reviewer 1 Report
Comments and Suggestions for Authors
Thank you very much for the opportunity to read this paper. The study of the determinants of health expenditures is interesting. I also like the microdata used. There are many different specifications and robustness checks. See many comments.
1)
Please highlight the theoretical and empirical contribution of your work better. Numerous studies have investigated the determinants of out of pocket health expenditures. The research questions can be sharpened. The introduction should outline: 1. What is already known about the topic? 2. What is unknown about the subject, and what does the study intend to examine? This means outlining the research gap you seek to fill. 3. What are the specific research questions the study focuses on?
2)
Please better justify the econometric method. OLS in first differences is not a good idea. Use the static, fixed effects estimator as a starting point. The first differenced GMM estimator developed by Arellano and Bond (1991) is outdated. You should consider to use the system GMM estimator developed by Blundell and Bond (1998) or dynamic ML estimator by Hsiao et al. 2002. However, you only have four years of data. So, the static fixed effects model is a good start.
3)
Can you link the microdata over the years? Often, there is a rotating survey. More information is needed. Do you have an ID number with which you can link the data? This is very important to enable replication. Is it an unbalanced panel data set?
4)
My other question relates to the definition of the dependent variables. Do you have zero observations in the dependent variables? It could be that there is no health expenditure. Many people don't spend anything on health care expenditures. If you have zero values, you need to use the proportion equation methods of fractional regression methods.
5)
Please provide descriptive statistics for all variables (means, standard deviation and min and maximum).
6)
Please interpret the magnitude of the relationships better and not only the direction and significance of the variables.
7)
Using ordinal categorical variables as explanatory variables is not a good idea. Use a set of dummy variables.
8)
The literature review needs to be improved. There are only 20 references. Some of the journals listed are of poor quality. There are many studies on the determinants of health expenditure.
Minor comments
a)
This estimator is outdated
Arellano M, Bond S. Some tests of specification for panel data: Monte Carlo evidence and an application to employment equa-463 tions. The Review of Economic Studies. 1991. 58(2):277–297.
Try this one
Blundell, R., & Bond, S. (1998). Initial conditions and moment restrictions in dynamic panel data models. Journal of econometrics, 87(1), 115-143.
This estimator works with four time periods:
Hsiao, C., Pesaran, M. H., & Tahmiscioglu, A. K. (2002). Maximum likelihood estimation of fixed effects dynamic panel data models covering short time periods. Journal of econometrics, 109(1), 107-150.
b)
Rewrite this sentence
This research paper fills this gap by producing unbiased and consistent estimates of the effects of health measured by activity limitations and chronic conditions on OOPEs across different subgroups among the middle-aged and older adults in China and explores potentially different effects of activity limitations and chronic conditions on OOPEs.
To be read as
This research paper examines the effects of health status, as measured by activity limitations and chronic diseases, on OOPEs in different subgroups among middle-aged and elderly adults in China. It also examines the potentially different effects of activity limitations and chronic diseases on OOPEs.
No abbreviations. Write out OOPEs
c)
Delete this paragraph. This reads like it was written by Chatgbt
The Materials and Methods should be described with sufficient details to allow others to replicate and build on the published results. Please note that the publication of your manuscript implicates that you must make all materials, data, computer code, and protocols associated with the publication available to readers. Please disclose at the submission stage any restrictions on the availability of materials or information. New methods and protocols should be described in detail while well-established methods can be briefly described and appropriately cited.
Start here:
Many studies have estimated health care expenditure and extra cost of disability across demographic and socioeconomics groups. Owens (2008) analyze the gender differences in health care expenditures and resource utilizations in United States. He finds that health care expenditures tend to be higher among women than among men. The greatest disparity in health care spending between men and women occurred in the population aged 45 to 64 years with women’s health issues primarily re-volving around chronic conditions and menopausal symptoms in this age group [8]. Loyalka et al (2014) measure the extra cost of disability across different types of disability and households in both urban and rural areas of China. They find the extra costs of disability are larger for urban households than rural households and there is a strong negative correlation between disability and household income [9].
d)
The language is not always smooth. The language must be improved.
Comments on the Quality of English LanguageEnglish is not acceptable
Author Response
1) Please highlight the theoretical and empirical contribution of your work better. Numerous studies have investigated the determinants of out of pocket health expenditures. The research questions can be sharpened. The introduction should outline: 1. What is already known about the topic? 2. What is unknown about the subject, and what does the study intend to examine? This means outlining the research gap you seek to fill. 3. What are the specific research questions the study focuses on?
Reply: Thank you very much for this important comment. I have outlined the introduction and revised the contributions as suggested.
2) Please better justify the econometric method. OLS in first differences is not a good idea. Use the static, fixed effects estimator as a starting point. The first differenced GMM estimator developed by Arellano and Bond (1991) is outdated. You should consider to use the system GMM estimator developed by Blundell and Bond (1998) or dynamic ML estimator by Hsiao et al. 2002. However, you only have four years of data. So, the static fixed effects model is a good start.
Reply: Thank you for this important comment. I have revised the model accordingly, beginning with the fixed effects estimator. Following this, I have adopted the system-GMM estimator developed by Blundell and Bond (1998), in preference to the difference-GMM, due to its enhanced robustness and efficiency.
3) Can you link the microdata over the years? Often, there is a rotating survey. More information is needed. Do you have an ID number with which you can link the data? This is very important to enable replication. Is it an unbalanced panel data set?
Reply: Thank you. Yes, the microdata can be linked over the years. This paper uses the 2011, 2013, 2015 and 2018 waves of China Health and Retirement Longitudinal Study (CHARLS), which is a longitudinal study of individuals over age 45 in China. This study focuses on 7,448 respondents aged 50 and over that can be followed over the four waves from 2011 to 2018, which forms a balanced panel data after removing missing values. More details of the data used in this paper can be referenced in ‘Data and Methods’ part and code files shared.
4) My other question relates to the definition of the dependent variables. Do you have zero observations in the dependent variables? It could be that there is no health expenditure. Many people don't spend anything on health care expenditures. If you have zero values, you need to use the proportion equation methods of fractional regression methods.
Reply: Thanks for the advice. Yes, OOPEs have zero observations in the dependent variables, meaning that there is no health expenditure. I have added one to the dependent variable ‘OOPEs’ in dollars and taken the logarithm to deal with zero values in data, which also helps in normalizing data and reducing the effect of outliers.
5) Please provide descriptive statistics for all variables (means, standard deviation and min and maximum).
Reply: Thank you. The descriptive statistics have been expanded as suggested.
6) Please interpret the magnitude of the relationships better and not only the direction and significance of the variables.
Reply: Thank you for your input. Could you please provide more specific suggestions to help me refine this interpretation further?
7) Using ordinal categorical variables as explanatory variables is not a good idea. Use a set of dummy variables.
Reply: Thank you. The dummy variables have been created for categorical explanatory variables. For more details, please refer to the shared code files.
8) The literature review needs to be improved. There are only 20 references. Some of the journals listed are of poor quality. There are many studies on the determinants of health expenditure.
Reply: Thank you. Additional references have been included.
Minor comments
a)
This estimator is outdated
Arellano M, Bond S. Some tests of specification for panel data: Monte Carlo evidence and an application to employment equa-463 tions. The Review of Economic Studies. 1991. 58(2):277–297.
Try this one
Blundell, R., & Bond, S. (1998). Initial conditions and moment restrictions in dynamic panel data models. Journal of econometrics, 87(1), 115-143.
This estimator works with four time periods:
Hsiao, C., Pesaran, M. H., & Tahmiscioglu, A. K. (2002). Maximum likelihood estimation of fixed effects dynamic panel data models covering short time periods. Journal of econometrics, 109(1), 107-150.
Reply: Thank you. Please see the answer to #2).
- b) Rewrite this sentence
This research paper fills this gap by producing unbiased and consistent estimates of the effects of health measured by activity limitations and chronic conditions on OOPEs across different subgroups among the middle-aged and older adults in China and explores potentially different effects of activity limitations and chronic conditions on OOPEs.
To be read as
This research paper examines the effects of health status, as measured by activity limitations and chronic diseases, on OOPEs in different subgroups among middle-aged and elderly adults in China. It also examines the potentially different effects of activity limitations and chronic diseases on OOPEs.
No abbreviations. Write out OOPEs
Reply: Thank you for pointing that out. This sentence has been rewritten to align with the introduction. Also, I’ve spelled out 'OOPEs' in full upon its first appearance in the text, using the abbreviation only subsequently.
- c) Delete this paragraph. This reads like it was written by Chatgbt
The Materials and Methods should be described with sufficient details to allow others to replicate and build on the published results. Please note that the publication of your manuscript implicates that you must make all materials, data, computer code, and protocols associated with the publication available to readers. Please disclose at the submission stage any restrictions on the availability of materials or information. New methods and protocols should be described in detail while well-established methods can be briefly described and appropriately cited.
Start here:
Many studies have estimated health care expenditure and extra cost of disability across demographic and socioeconomics groups. Owens (2008) analyze the gender differences in health care expenditures and resource utilizations in United States. He finds that health care expenditures tend to be higher among women than among men. The greatest disparity in health care spending between men and women occurred in the population aged 45 to 64 years with women’s health issues primarily re-volving around chronic conditions and menopausal symptoms in this age group [8]. Loyalka et al (2014) measure the extra cost of disability across different types of disability and households in both urban and rural areas of China. They find the extra costs of disability are larger for urban households than rural households and there is a strong negative correlation between disability and household income [9].
Reply: Thank you for your attention to detail. The paragraph was inadvertently included from the journal template; it has now been removed.
Comments on the Quality of English Language
The language is not always smooth. The language must be improved. English is not acceptable
Reply: I have made some edits throughout.
Reviewer 2 Report
Comments and Suggestions for Authors
The article is dedicated to an urgent topic and investigates the issues of out-of-pocket health expenditures and health indicators of the population in China. The paper is interesting from a practical point of view, especially for understanding how to optimize the financial topics of the healthcare sector functioning and maintaining its most tremendous components. As for the pluses of the paper, it is well-structured and comprises a deep literature review on the studied topic. The paper covers some scientific and practical healthcare management gaps, revealing valuable relationships between health problems and OOPEs. The manuscript is scientifically sound and gives appropriate and understandable results for its reader.
Besides, here are some aspects that must be considered by the authors before manuscript acceptance:
1) Take care of all abbreviations in the manuscript. Some of them are explained too late in the text or are not explained but are in the abstract – for example, OOPEs.
2) Keywords. I recommend changing the words “Social Protection” to “Social health insurance.” It is more appropriate considering the main text of the article.
3) Lines 57-63: seems like they are redundant.
4) Lines 87-89: the repeating of the text, which was in the previous section (lines 47-49).
5) Line 119. The author should explain the reason for choosing that category of age for respondents (why aged 50 and over?)
6) Subsection 3.2. Why did the author take into account five tasks for IADL, not six as it is for ADL to balance them? Explain in the text.
7) Line 165. Add an explanation of the approach to the normalization of indexes.
8) Do not use in the text “my”, “I do,” “I start,” “I estimate” and similar words and phrases as it is not in a scientific manner. Try to avoid them and use phrases written from the third person.
Author Response
The article is dedicated to an urgent topic and investigates the issues of out-of-pocket health expenditures and health indicators of the population in China. The paper is interesting from a practical point of view, especially for understanding how to optimize the financial topics of the healthcare sector functioning and maintaining its most tremendous components. As for the pluses of the paper, it is well-structured and comprises a deep literature review on the studied topic. The paper covers some scientific and practical healthcare management gaps, revealing valuable relationships between health problems and OOPEs. The manuscript is scientifically sound and gives appropriate and understandable results for its reader.
Besides, here are some aspects that must be considered by the authors before manuscript acceptance:
1) Take care of all abbreviations in the manuscript. Some of them are explained too late in the text or are not explained but are in the abstract – for example, OOPEs.
Reply: Thank you for pointing that out. I’ve spelled out all abbreviations in full upon its first appearance in the text, using the abbreviation only subsequently.
2) Keywords. I recommend changing the words “Social Protection” to “Social health insurance.” It is more appropriate considering the main text of the article.
Reply: Thank you. I’ve changed the words “Social Protection” to “Social Health Insurance” as recommended.
3) Lines 57-63: seems like they are redundant.
Reply: Thank you for your attention to detail. The paragraph was inadvertently included from the journal template; it has now been removed.
4) Lines 87-89: the repeating of the text, which was in the previous section (lines 47-49).
Reply: Thank you. The previous section has been rephrased to avoid repetitions.
5) Line 119. The author should explain the reason for choosing that category of age for respondents (why aged 50 and over?)
Reply: Thank you. I’ve added the explanations in the context:
“This paper uses the 2011, 2013, 2015 and 2018 waves of China Health and Retirement Longitudinal Study (CHARLS), which is a nationally representative longitudinal survey of people aged 45 and over and their partners in China.”
“Since ADL/IADL related questions were not asked of respondents aged 45-49, this study focuses on 7,448 respondents aged 50 and over who can be followed over the four waves from 2011 to 2018, forming a balanced panel data set after removing missing values.”
6) Subsection 3.2. Why did the author take into account five tasks for IADL, not six as it is for ADL to balance them? Explain in the text.
Reply: Disability was measured using difficulties in activities of daily living questions. These questions were initially developed to capture the physical effects of aging. In some surveys, the questions are only administered to respondents above a certain age (e.g. 50 and above in CHARLS). Since the 1980s, they have been used in a variety of clinical, policy, and research contexts (Jette, 1994; Mathiowetz & Lair, 1994; Stewart and Ware, 1992; Wiener, Hanley, Clark, & Van Nostrand, 1990).
There are different types of activities of daily living questions, including basic and instrumental. The first measure, ADLs, captures basic ADLs that are fundamental for body functioning (e.g. walking a specific distance) and include self-care activities such as feeding oneself, going to the bathroom without help, and dressing without help, bathing, eating, walking, toileting, urination and defecation. The second measure, instrumental ADLs (IADLs), covers more complex tasks such as managing money.
More references are included in the text.
Jette, A. M. (1994). How measurement techniques influence estimates of disability in older populations. Social Science and Medicine, 38, 937-942.
Mathiowetz, N. A., & Lair, T. J. (1994). Getting better? Change or error in the measurement of functional limitations. Journal of Economic and Social Measurement, 20, 237-262.
Stewart, A., & Ware, J. (Eds.). (1992). Measuring functioning and well-being. Duke University Press.
Wiener, J. M., Hanley, R. J., Clark, R., & Van Nostrand, J. F. (1990). Measuring the activities of daily living: Comparisons across national surveys. Journal of Gerontology: Social Sciences, 45, S229-S237.
7) Line 165. Add an explanation of the approach to the normalization of indexes.
Reply: To capture the number and severity of ADL or IADL limitations, we use ADL and IADL indices as developed by Stewart and Ware (1992; page 80) and use in the literature since (e.g. Gertler and Gruber (2002)). The ADL index (from 0 to 100) is the normalized index of the sum of the answers (each equal to 1 (No, I don’t have any difficulty), 2 (I have difficulty but can still do it), 3 (Yes, I have difficulty and need help) or 4 (I cannot do it)) to the six ADL questions as follows with a minimum of six and a maximum of 24:
ADL Index = (SumADL - 6)/(24 - 6)x100
For instance, if someone answers 1 (No, I don’t have any difficulty) to the six ADL questions, the sum of answers (SumADL) is six and the ADL index is zero. If someone answers 1 (No, I don’t have any difficulty) to five ADL questions but a 4 (I cannot do it) to one ADL question, the sum of answers (SumADL) is 9 and the score is (9-6)/(24-6)x100=1/6x100
Similarly, the IADL index is the normalized score or sum of the answers (each equal to 1, 2, 3, or 4) to the five IADL questions as follows with a minimum of five and a maximum of 20: IADL Index = (SumIADL - 5)/(20 - 5)x100
Gertler, P., & Gruber, J. (2002). Insuring consumption against illness. American
Economic Review, 92, 51–70.
8) Do not use in the text “my”, “I do,” “I start,” “I estimate” and similar words and phrases as it is not in a scientific manner. Try to avoid them and use phrases written from the third person.
Reply: Thank you. I have replaced them and used phrases written from the third person.
Reviewer 3 Report
Comments and Suggestions for Authors
Good work but needs major revision

Author Response
Population ageing and related diseases in the mid age have been a crucial issue to the policy makers in the western economies as a particular scenario. It has also spread to the countries like Japan and China from the east. India and Brazil like countries are waiting to be get trapped under such a social and economic crisis. Keeping these facts in mind, the authors have attempted a good piece of work on the relationships between health effects upon out-of-pocket expenditure in China using the CHARLS and primary databases. The study has used statistical and econometric tools for reaching the results and their analyses. The results are expected as there are so many studies in the similar areas. It has some degrees of policy implications so far as the importance of mitigating the ageing issue is concerned. However, the study has some lacunae in some of the areas as mentioned below and the rectifications/revisions of which may be leading to a good research outcome.
- The Abstract should not contain any unidentified terms like OOPEs, they should be written in full unless otherwise stated earlier.
Reply: Thank you for pointing that out. I’ve spelled out all abbreviations in full upon its first appearance in the text, using the abbreviation only subsequently.
- The study has a brief introduction to the subject area. It starts with the issue of the health due to ageing but it has missed the real grounds behind such a crisis. The author should have focused, besides the existing ones, why such ageing and health issues are arising today. The factors, no other than pollution and urbanization have led to the accentuation and aggravation of the problems. The authors are thus suggested to incorporate these two issues into the Introduction part and add some related literatures on the topics to revise the Introduction and Literature Review sections. Here are some suggested references which the authors may use to strengthen the areas of concerns [Das R. C. Chatterjee. T. & Ivaldi E. (2023). Co-movements of income and urbanization through energy use and pollution: An ; Das, R. C. & Ivaldi, E. (2021): Is pollution a cost to health? Theoretical and empirical inquest for the Public Health, 18(12)]
Reply: Thank you for providing the literature. This study has talked about aging issue as a starting point. China’s aging problem is primarily driven by the historical effects of the one-child policy, which significantly lowered the birth rate, combined with increased life expectancy due to advancements in healthcare and living standards. Urbanization and modernization have also altered traditional family structures, leading to reduced familial support for the elderly. Then “The high prevalence of disability and chronic conditions among the old presents challenges to the public health system in China.”
This paper examines the dynamic relationship between health and out-of-pocket expenditures (OOPEs) among middle-aged and older adults in China. While urbanization and pollution could serve as useful control variables in such an analysis, this study utilizes the 2011, 2013, 2015, and 2018 waves of the China Health and Retirement Longitudinal Study (CHARLS). CHARLS is a nationally representative longitudinal survey targeting individuals aged 45 and above and their spouses. However, due to the scope of data collected by CHARLS, which is limited to demographic, socioeconomic, and health information at the individual and household levels, this paper does not include urbanization and pollution data in the model.
- Before going to the regression analysis, in a pooled data format, a good choice of the estimation process, the authors could present the correlation among the variables to see whether there is any sort of associations. It may be incorporated after the first table of descriptive statistics.
Reply: Thank you for the advice. There have been extensive studies that examine the correlations between OOPEs and health problems. Specifically, the correlation among the variables have been examined in my previous paper (Gao et al, 2023). This is the reason why I start with modeling in this paper with the purpose of generating unbiased and consistent estimates of the effects of health status on OOPEs.
Gao J, Kim H, Mitra S. Out-of-Pocket Health Expenditures Associated with Chronic Health Conditions and Disability in China. Int J Environ Res Public Health. 2023 Jul 27;20(15):6465. doi: 10.3390/ijerph20156465. PMID: 37569006; PMCID: PMC10418713.
- Why the authors have considered only the mentioned controlling only? They should incorporate the factors such as urbanization and environmental pollution to have better and more robust result of the impact of health issues upon the OOPEs. It is suggested to incorporate.
Reply: Thank you very much. Please see the answer to #2.
- Using a difference model is a good one to get robust results to avoid econometric issues. As the authors have considered the dynamic panel data, I would suggest focusing on the results more accurately. They could have used the dynamic panel estimation to find the lagged
Reply: Thank you. I have revised the model accordingly, beginning with the fixed effects estimator. Following this, I have adopted the system-GMM estimator developed by Blundell and Bond (1998), in preference to the difference-GMM, due to its enhanced robustness and efficiency.
- It is not clear about the motivations behind the study? All have no clear explanation; it is suggested to incorporate the same. Further, the study should categorically mention the Research Gaps to justify its contributions to the scientific literature as it used the data and results of so many other authors.
Reply: Thank you very much for this important comment. I have outlined the introduction with the motivations and revised the research gaps and contributions.
- The study did not present any theoretical basics of the interrelationships between the two key indicators without having any model. It is suggested to form a basic model with at least linking the two variables in equational structure to give a break to the monotonicity in the presentations.
Reply: Thank you very much. Please see the answer to #3.
- In writing the article, the author has used my aim, my study etc. which should be removed and all should be written in passive voices like the The study uses… etc.
Reply: Thank you. I have replaced them and used phrases written from the third person.
- The author should also discuss the policy formulation in further detail and should be all inclusively analyzed in the Discussion part.
Reply: Thank you very much. The policy formulation had been expanded in detail in the Discussion part.
Moderate revisions recommended…………..
Reply: I have made some edits throughout.
Academic Editor Notes
Some additional comments in addition to those of the reviewers, and note one reviewer has provided details of what should be changed in a separate file.
The author has also left in Author instructions from the Manuscript template that need deleting see lines 57-62 "The Materials and Methods should be described with sufficient details to allow others to replicate and build on the published results.
Reply: Yes, the paragraph was inadvertently included from the journal template; it has now been removed.
Please note that the publication of your 58 manuscript implicates that you must make all materials, data, computer code, and protocols associated with the publication available to readers. Please disclose at the submission stage any restrictions on the availability of materials or information. New methods and 61 protocols should be described in detail while well-established methods can be briefly described and appropriately cited."
Reply: Thanks for the reminder. I will make all the relevant materials including data and code available to readers.
Could the author elaborate on how they got approvals and then access to the CHARLS datasets.
Reply: CHARLS is a public dataset. Anyone can register and download the data.
Could the author also explain the reference year for costs and if these were adjusted to a single reference year?
Reply: The costs were adjusted to reference year 2015. RMB was converted to US dollars using the 2015 average exchange rate of 1 USD=6.28 RMB.
This manuscript has been submitted by a single author however in the methods it refers to "we" and "our" implying there was more than one contributor for this research see " ...we multiplied the value by 12 to estimate annual outpatient expenditures. OOPEs, our target" (lines 151-152). there are also many other examples throughout the paper. Please explain why this is a single-author paper or please ensure co-authors are added as appropriate, or name the research support people in an Acknowledgments section.
Reply: "we" and "our" should be replaced by “I” and “my”, or use phrases written from the third person as suggested by reviewers. This is a single-author paper completed for one of my PhD dissertations, but my supervisor did give me a lot of advice on it. I’ve added the name of my supervisor in the Acknowledgements section. May I ask my supervisor whether she would like to be included as a co-author should this paper be accepted?
Round 2
Reviewer 1 Report
Comments and Suggestions for Authors
Thank you for the revision. You have provided new estimates, which I can see are dynamic panel data estimates. However, you need to explain the model's specification and the different explanatory variables better.
Please carefully motivate the variables ADL index, IADL index, and NCD index. I cannot find descriptive statistics for them.
And they should be theoretically motivated.
The socio-economic characteristics should also be included; however, they are time-invariant. What about income?
Also, more information about the data set is needed. Can you link the different waves over time? Is it an unbalanced panel or an unbalanced panel? How have you linked the data over time? You need to interpret the magnitude of the relationships better, not just the significance and direction. The data ends in 2018. Are there more recent waves?
minor comments
a)
Please look at the numbering
3.2. Out-of-pocket expenditures (OOPEs) 162
3.2. Health Measures
b)
Table 2. Descriptive statistics
What are the units?
Percentages?
c)
Table 3-1. First difference regression of the effects of health on the natural logarithm of OOPEs, stratified by gender and income
Can you provide more details about the regression method? Is it OLS based on the first differences of the variables?
The fixed effects model is preferred.
Comments on the Quality of English LanguageI can look at the language when I receive the revised version.
Author Response
Thank you for the revision. You have provided new estimates, which I can see are dynamic panel data estimates. However, you need to explain the model's specification and the different explanatory variables better.
Reply: Thank you very much. The details of the model specification are provided when introducing the fixed effects model (lines 207-215). This specification also applies to the GMM model.
Please carefully motivate the variables ADL index, IADL index, and NCD index. I cannot find descriptive statistics for them. And they should be theoretically motivated.
Reply: Thank you. I’ve created a new descriptive table (Table 2) to summarize the descriptive statistics showing the association between ADL, IADL, and Multimorbidity status and OOPEs.
Also, to capture the number and severity of ADL, IADL limitations and chronic health conditions, I use ADL and IADL indices as developed by Stewart and Ware (1992; page 80) and use in the literature since (e.g. Gertler and Gruber (2002)). The ADL index (from 0 to 100) is the normalized index of the sum of the answers (each equal to 1 (No, I don’t have any difficulty), 2 (I have difficulty but can still do it), 3 (Yes, I have difficulty and need help) or 4 (I cannot do it)) to the six ADL questions as follows with a minimum of six and a maximum of 24:
ADL Index = (SumADL - 6)/(24 - 6)x100
For instance, if someone answers 1 (No, I don’t have any difficulty) to the six ADL questions, the sum of answers (SumADL) is six and the ADL index is zero. If someone answers 1 (No, I don’t have any difficulty) to five ADL questions but a 4 (I cannot do it) to one ADL question, the sum of answers (SumADL) is 9 and the score is (9-6)/(24-6)x100=1/6x100
The IADL index is the normalized score or sum of the answers (each equal to 1, 2, 3, or 4) to the five IADL questions as follows with a minimum of five and a maximum of 20: IADL Index = (SumIADL - 5)/(20 - 5)x100
Gertler, P., & Gruber, J. (2002). Insuring consumption against illness. American
Economic Review, 92, 51–70.
Similarly, a measure of chronic health conditions that captures 15 chronic conditions (hypertension, diabetes, cancer, chronic lung disease, heart disease, stroke, psychiatric problem, arthritis, dyslipidemia, liver disease, kidney disease, digestive disease, asthma, depression and memory problem) is used. Each respondent answered the question regarding whether or not they have been diagnosed by the doctor for any of the conditions above. Then the number of diagnosed chronic conditions for each respondent is counted. A Non-Communicable Disease (NCD) index is constructed by normalizing the count to 100.
I’ve added some explanations and references in Health Measures Section.
The socio-economic characteristics should also be included; however, they are time-invariant. What about income? Also, more information about the data set is needed. Can you link the different waves over time? Is it an unbalanced panel or an unbalanced panel? How have you linked the data over time? You need to interpret the magnitude of the relationships better, not just the significance and direction. The data ends in 2018. Are there more recent waves?
Reply: Thank you. Income is time variant.
The microdata can be linked over the years. This paper uses the 2011, 2013, 2015 and 2018 waves of China Health and Retirement Longitudinal Study (CHARLS), which is a longitudinal study of individuals over age 45 in China. This study focuses on 7,448 respondents aged 50 and over that can be followed over the four waves from 2011 to 2018, which forms a balanced panel data after removing missing values. In other words, the same individuals were surveyed, and data were collected during each wave. I’ve added this information in the context. More details of the data used in this paper can be referenced in ‘Data and Methods’ part and code files shared.
Regarding the magnitude of the relationships, comparisons were made showing that an increase in the NCD index leads to a higher rise in OOPEs compared to the ADL/IADL index. Additionally, comparisons have been made showing that increases in health measures lead to a larger rise in OOPEs for the Urban Hukou/Urban Residence group compared to the Rural Hukou/Rural Residence group. Changes in health measures and OOPEs over the years could be analyzed using a 'split first difference' model by focusing separately on negative changes and positive changes. However, this analysis is beyond the scope of the current study.
The more recent waves have not been updated. I guess the CHARLS is still working on it. The results shall be updated if the more recent wave is available, which will give more robust results. I also mentioned it in the limitation part.
minor comments
- a) Please look at the numbering
3.2. Out-of-pocket expenditures (OOPEs) 162
3.2. Health Measures
Reply: Thanks for pointing out. I have revised it.
- b) Table 2. Descriptive statistics: What are the units? Percentages?
Reply: Thanks. I’ve updated the descriptive statistics in Table 2 to display percentages instead of two-decimal figures.
- c) Table 3-1. First difference regression of the effects of health on the natural logarithm of OOPEs, stratified by gender and income Can you provide more details about the regression method? Is it OLS based on the first differences of the variables? The fixed effects model is preferred.
Reply: Thank you. Equation 1 is fixed effect model. I’ve added more explanations in the context: “The first-difference estimator addresses the issue of individual fixed effects by using the changes between two time periods for each individual. This approach effectively eliminates the fixed individual-specific effects, as these do not vary over time. The model specification for this estimator is as follows…”
Reviewer 3 Report
Comments and Suggestions for Authors
minor revision required
Author Response
Thank you. I’ve read the recommended references carefully. For the first one (Das R. C. Chatterjee. T. & Ivaldi E.2023), it talks about the associations between the four terms: income, urbanization, energy uses and GHG emissions. The paper provides evidence on 1) economic growth as driving forces of pollution/carbon emission; 2) the impact of rapid urbanization on pollution. And in this section, the author introduces one literature related to health in China: “Fang et al. (2015) establish the adverse effect of urbanization on pollution in terms of rapid increase in surface temperature on Earth, and also claims adverse effect on human health following the same reason.”; 3) the unidirectional relationship between green energy on carbon emission as well as the negative effects of urbanization on carbon emission; 4) urbanization increases energy consumption, which causes negative effects on environment.
For the second reference (Das, R. C. & Ivaldi, E., 2021), it talks about the common phenomenon in developing economies in the short run stage of Environmental Kuznets Curve, rising income and rising health expenditure may lead to rising pollution. The results show that in the short run, pollution is the cause of health expenditures for many developed countries, and health expenditures are the cause of pollution in some of the developing countries.
However, my research is focusing on the effect of health on OOPEs (healthcare cost not covered by the health insurance) in China. As for the research design, I referenced tons of previous research in this topic (Owens 2008; Loyalka et al 2014; Lee et al 2014; Lee et al 2016; Meraya et al 2017, Salinas-Rodriguez et al 2020, Kim and Mitra 2022, etc.). Pollution and urbanization might be relevant, but they are not the study object of this research. Also, I mentioned aging issue since the paper is talking about the effect of health on OOPEs among the middle-aged and older adults. To conclude, I find it quite challenging to incorporate pollution and urbanization into this research. I believe they are beyond the scope of this research, as the use of CHARLS doesn’t align with these aspects as well.